CYP3A4 and CYP3A5: the crucial roles in clinical drug metabolism and the significant implications of genetic polymorphisms

Zhang Yuqing 1 2
Wang Ziying 2
Wang Yuchao 2
Jin Weikai 2
Zhang Zheyan 2
Jin Lehao 2
Qian Jianchang 1 2 qianjc@wmu.edu.cn
Zheng Long 1 7270684@qq.com
1 Affiliated Yueqing Hospital, Wenzhou Medical University , Wenzhou, Zhejiang , China
2 Institute of Molecular Toxicology and Pharmacology, School of Pharmaceutical Sciences, Wenzhou Medical University , Wenzhou, Zhejiang , China
Uversky Vladimir
Electronic publication date: 2024 Dec 5
Publication date: 2024
Volume: 12
Electronic Location ID: e18636
Received 2024 Aug 8; Accepted 2024 Nov 12
Copyright: © 2024 Zhang et al.
Copyright year: 2024
Copyright holder: Zhang et al.
License: This is an open access article distributed under the terms of the Creative Commons Attribution License, which permits unrestricted use, distribution, reproduction and adaptation in any medium and for any purpose provided that it is properly attributed. For attribution, the original author(s), title, publication source (PeerJ) and either DOI or URL of the article must be cited.
License URL: https://creativecommons.org/licenses/by/4.0/

Keywords: CYP3A4, CYP3A5, Genetic polymorphism, Drug interactions, Inter-individual variability, Enzyme kinetics, Drug metabolism

Funding: The authors received no funding for this work.

==============================
CYP3A, a key member of the cytochrome P450 (CYP450) superfamily, is integral to drug metabolism, processing a substantial portion of medications. Their role in drug metabolism is particularly prominent, as CYP3A4 and CYP3A5 metabolize approximately 30–50% of known drugs. The genetic polymorphism of CYP3A4/5 is significant inter-individual variability in enzymatic activity, which can result in different pharmacokinetic profiles in response to the same drug among individuals. These polymorphisms can lead to either increased drug toxicity or reduced therapeutic effects, requiring dosage adjustments based on genetic profiles. Consequently, the study of the enzymatic activity of CYP3A4/5 gene variants is of great importance for the formulation of personalized treatment regimens. This article first reviews the role of CYP3A4/5 in drug metabolism in the human body, including inhibitors and inducers of CYP3A4/5 and drug-drug interactions. In terms of genetic polymorphism, it discusses the detection methods, enzymatic kinetic characteristics, and clinical guidelines for CYP3A5. Finally, the article summarizes the importance of CYP3A4/5 in clinical applications, including personalized therapy, management of drug-drug interactions, and adjustment of drug doses. This review contributes to the understanding of the functions and genetic characteristics of CYP3A4/5, allowing for more effective clinical outcomes through optimized drug therapy.

Introduction

Cytochrome P450 (CYP450) is the most significant phase I drug metabolizing enzyme in the human body, and approximately 75–90% of commonly utilized clinical drugs are metabolized by CYP450 (Lee et al., 2024; Zhou, Liu & Chowbay, 2009). Currently, 57 CYP-encoding genes and 58 pseudogenes are known to exist on human chromosomes, which can be further divided into 18 families and 43 subfamilies based on amino acid sequence similarity (Zhao et al., 2021b). The CYP3 family includes the CYP3A subfamily, consisting of four genes located on chromosome 7: CYP3A4, CYP3A5, CYP3A7, and CYP3A43. CYP3A4 is a key drug-metabolizing enzyme in the CYP450 superfamily and is the most abundantly expressed drug-metabolizing enzyme in the human liver (about 95% of them) (Martiny & Miteva, 2013), accounting for an average of 14–24% of the total microsomal CYP450 pool in vivo (Shimada et al., 1994). They are involved in the pre-systemic and systemic metabolism of 30–50% of the clinical drugs (Zanger & Schwab, 2013) because their active sites are large and sufficiently flexible to bind and metabolize many relatively structurally large lipophilic compounds (Williams et al., 2000). CYP3A4/5 are localized in the cellular endoplasmic reticulum and are induced to be expressed by glucocorticoids and some drugs (Miedziaszczyk & Idasiak-Piechocka, 2023; Rojas Velazquez, Noebauer & Pandey, 2022). The apparent structural diversity of the substrates of this enzyme, including drugs of different molecular sizes and pharmacological effects, has made it an important focus for the study of drug metabolism in vivo. CYP3A4/5 play a crucial role in the metabolism of numerous drugs in the body. They facilitate various metabolic transformations, such as N-oxidation, C-oxidation, N-dealkylation, O-dealkylation, nitro-reduction, demethylation, dehydration, and C-hydroxylation (Bieler et al., 2003; Rendic, 2021; Yue & Hirao, 2023). Variations in CYP3A4/5 activity among inter-individuals are largely due to factors like age, gender, daily routines, medication use, and health conditions (Gloor et al., 2022; Huang et al., 2024). In contrast, intra-individual differences are mainly due to the generation of new variants by single nucleotide gene polymorphisms (SNPs) in the corresponding coding regions of CYP3A4/5. SNPs can affect multiple gene steps like transcription, translation, and protein synthesis. This then affects the biological expression level of CYP3A4/5, resulting in differences in the metabolic efficiency of various drugs among individuals. For example, CYP3A4*8 (rs35599367), CYP3A4*11 (rs4986910), CYP3A4*17 (rs776746), CYP3A5*3A (rs776746), etc., may lead to a decrease in CYP3A4 enzyme activity (Eap et al., 2004; García-Martín et al., 2002; Oliver, Lubomirov & Carcas, 2010), which results in a decrease in the metabolism rate of the drug, an extension of the retention time in the body, elevated or reduced drug efficacy, and increased risk of drug toxicity and side effects. On the contrary, another part of the variant CYP3A4*18A (rs2740574) may have an increased activity of the CYP3A4 enzyme (Lepper et al., 2005), and the body metabolizes the drug faster, which leads to a decrease in the efficacy of the drug. Changes in enzyme activity may affect drug efficacy and safety, so understanding the function of these polymorphic sites can be used to guide the selection and adjustment of dosage of drugs metabolized by CYPA4/5, which is essential for achieving individualized drug therapy. The review focusing on CYP3A4 and CYP3A5 is intended for an audience encompassing medical professionals, researchers, and students with an interest in pharmacogenomics, drug metabolism, and personalized medicine. It also caters to pharmaceutical scientists and regulatory authorities involved in drug development and policy-making, who require a deep understanding of genetic variations’ impact on drug efficacy and safety.

Survey methodology

To ensure a comprehensive and systematic literature review on the role of CYP3A4 and CYP3A5 in drug metabolism, we conducted a thorough search across various databases including PubMed (https://pubmed.ncbi.nlm.nih.gov/), Web of Science (https://www.webofscience.com), and CNKI (https://www.cnki.net). Our search strategy involved the use of both English and Chinese keywords related to the enzymes’ genetic polymorphisms, their impact on drug metabolism, and associated clinical implications. The keywords encompassed terms such as CYP3A4, CYP3A5, genetic polymorphism, drug metabolism, and personalized medicine, along with specific drug names and concepts relevant to our study. We systematically searched for studies that investigated the role of CYP3A4/5 in drug metabolism. During the screening process, we applied specific exclusion criteria to ensure the relevance and quality of the data included in our review. Studies that did not directly address CYP3A4/5’s role in drug metabolism or lacked clinical data were excluded from this review. The search was refined to include only the most relevant and recent articles by removing duplicates, assessing the availability of full texts, and prioritizing literature with a significant impact factor. We excluded studies that did not specifically address the therapeutic effects or the role of CYP3A4 and CYP3A5 in the context of drug metabolism. After a meticulous evaluation process, we selected 153 articles that met our inclusion criteria, focusing on the most current and influential research to provide a solid evidence base for our review.

Structure and function of CYP3A4 and CYP3A5

The CYP3A4 gene is located in the 7q21-22.1 region of chromosome 7 and consists of 13 exons and 12 introns, with a total length of approximately 27 Kbp, encoding 503 amino acid residues. The CYP3A5 gene is also located on chromosome 7, specifically in the 7q22.1 region, and consists of 13 exons and 12 introns, with a total length of approximately 62 Kbp, encoding approximately 441 amino acid residues (Lamba et al., 2002). CYP3A4/5 are mainly found in the liver and small intestine and are the main sites of drug metabolism (Lamba et al., 2002; Zeigler-Johnson et al., 2004). CYP3A4 features a phenylalanine cluster, which forms a substrate-binding pocket with the active site. Phenylalanine residues in this cluster can influence CYP3A4 activity, and conformational changes can increase the active site’s volume, allowing for multiple substrate binding (Kaur et al., 2016). A homology model of CYP3A4 identified key residues affecting substrate binding and metabolism (Williams et al., 2004). CYP3A5, while structurally similar to CYP3A4, has a higher and narrower active site, which contributes to its higher selectivity for certain substrates like schisantherin E (Wright, Chenge & Chen, 2019). Conformational differences in the helix F-G region further distinguish CYP3A5 (Hsu & Johnson, 2019; Wang et al., 2021). CYP function is characterized by a buried active site, a heme iron-cysteine bond crucial for redox states during catalysis, and the necessity of two synchronized electron-transfer steps for catalytic cycling (Urban et al., 2018). As shown in Figs. 1B and 1C, both CYP3A4 and CYP3A5 can bind to clotrimazole (Figs. 1A–1C). The overall structural framework of CYP3A4 and CYP3A5 is similar, both containing multiple α-helix and β-folding domains with very similar folding modes, and can basically overlap (Fig. 1D). In substrate binding, there are specific regions and preliminary recognition energies. Several crystal structures of CYP3A4 have been solved, revealing a ligand-accessible volume of the binding pocket estimated to be as large as 520 Å3 (Wright, Chenge & Chen, 2019). This allows for the binding of a wide variety of ligands and the adoption of multiple conformations. The binding pocket of CYP3A4 appears shorter and more horizontal compared to CYP3A5, whose pocket has a higher and narrower roof. These structural nuances are likely responsible for the observed selectivity in ligand binding and metabolism. CYP3A4 and 3A5 metabolize a broad range of substrates due to high sequence similarity (83% homology) and a large, promiscuous binding pocket (De Wildt et al., 1999). This allows for the metabolization of diverse compounds, including immunosuppressants, macrolide antibiotics, anticancer drugs, benzodiazepines, HMG-CoA reductase inhibitors, and anaesthetics. CYP3A4 also metabolizes the antifungal drug terbinafine through a complex N-dealkylation pathway (Maggo, Kennedy & Clark, 2011; Pratt et al., 2023; Shang et al., 2024). Although less abundant, CYP3A5 significantly contributes to drug metabolism and shows superior metabolism for certain drugs compared to CYP3A4 (Li et al., 2024; Seligson et al., 2023; Tseng et al., 2018). Both enzymes are potent steroid hydroxylases, metabolizing endogenous steroids and bile acids (Willson & Kliewer, 2002). Midazolam, triazolam, testosterone, and nifedipine are CYP3A metabolism probes in vitro (Mao et al., 2017). However, phenotypic data for different substrates often do not correlate, reflecting CYP3A4’s variable enzyme activity regulation and multiple overlapping substrate-binding regions (Takano et al., 2016).

Figure 1 The distribution and metabolism of CYP3A4 and CYP3A5.

(A) CYP3A4 and CYP3A5 are localized in the cellular photoplasmic endoplasmic reticulum. (B) CYP3A4 (green) cocrystallized with single clotrimazole (PBD: 8SPD). (C) CYP3A4 (blue) cocrystallized with clotrimazole (PBD: 8SG5), similar to CYP3A4. (D) Align (B) and (C), CYP3A4 and CYP3A5 overlap significantly. (E) CYP3A4 cocrystallizes with clotrimazole, and a single clotrimazole (blue in the center) binds to the heme iron (red in the center) in CYP3A4, limiting the space above the heme and expanding the active site cavity. (F) Various P450 isoforms and percentage of clinically used drugs metabolized by these isoforms. (G) Relative amounts of human hepatic CYP450 proteins. Created in BioRender.

Relative contribution and impact of genetic polymorphism of CYP3A4 and CYP3A5

CYP3A4 is the most abundant CYP3A enzyme in the liver and is responsible for the metabolism of approximately 30–50% of clinically used drugs. Its broad substrate specificity allows it to handle a wide range of structurally diverse compounds. In contrast, CYP3A5, which shares approximately 83% sequence homology with CYP3A4, exhibits a more limited tissue distribution and a narrower substrate preference. However, in individuals who are CYP3A5 expressors (those lacking the CYP3A5*3), this enzyme can significantly contribute to the metabolism of certain drugs, such as tacrolimus and nifedipine (Choong et al., 2024; Hou et al., 2024). Genetic polymorphisms within the CYP3A4 and CYP3A5 genes can lead to the production of enzyme variants with altered catalytic properties. For example, the CYP3A4*22 has been associated with reduced enzyme activity, which may result in altered drug clearance and potential drug-drug interactions (Collins & Wang, 2020). The CYP3A5*3, which is prevalent in many populations, leads to a nonfunctional enzyme due to a premature stop codon (García-Roca et al., 2012). This polymorphism is a key determinant of whether an individual will be an expressor or nonexpressor of CYP3A5 activity. Thus, understanding the interplay between CYP3A4 and CYP3A5, and how genetic variants influence their activities, is crucial for predicting drug metabolism and for the development of personalized treatment regimens.

Regulation of CYP3A4 and CYP3A5

CYP3A4/5, as important metabolic enzymes, assume key roles in the breakdown of many drugs, hormones, and other compounds in the human body. The activities of these enzymes are regulated and induced by a variety of factors, which in turn affect their functions in drug metabolism and drug-drug interactions. An in-depth understanding of the regulation and induction mechanisms of CYP3A4/5 is important for individualized drug therapy and prevention of drug-drug interactions.

Transcriptional regulation and post-transcriptional regulation

In cells, CYP3A4/5, as part of the cytochrome P450 enzyme family, metabolizes diverse compounds. Precise control mechanisms include transcriptional and post-transcriptional regulation, adjusting enzyme activities to physiological and environmental conditions by modulating gene expression and protein stability.

Transcriptional regulation

Transcriptional regulation of CYP3A4/5 is vital in drug metabolism, acting alongside post-transcriptional mechanisms. It involves transcription factor binding to DNA, promoter methylation, and chromatin conformation changes, impacting gene expression and cellular enzyme levels. Understanding this regulation enhances drug metabolism comprehension. The Pregnane X Receptor (PXR) modulates transcription by binding endogenous and exogenous compounds, thereby regulating detoxification pathways. PXR binds response elements, activating or repressing CYP3A4/5 transcription. PXR binds to response elements, activating or repressing the transcription of CYP3A4/5. A study detects the effect of PXR agonists on CYP3A4 expression through transient transfection of reporter genes. This method involves connecting PXR response elements to reporter genes (such as the luciferase gene) and transfecting them into cells to evaluate the transcriptional regulatory activity of PXR by detecting the expression level of the reporter genes (Luo et al., 2002). Similarly, Hepatocyte Nuclear Factor 4-alpha (HNF4α) directly regulates CYP3A4/5 transcription. Epigenetic modifiers enhance HNF4α expression, improving drug metabolism (Ruoß et al., 2019). A previous study included a detailed description of how HNF4α interacts with the CYP3A4/5 promoter using chromatin immunoprecipitation (ChIP) assays and how promoter methylation status is analyzed using bisulfite sequencing (Surapureddi, Rana & Goldstein, 2011). In addition to PXR and HNF4α, other transcription factors are also involved in the regulation of CYP3A4/5. For example, the Constitutive Androstane Receptor (CAR) maintains CYP3A4/5 levels in the liver, which is crucial for drug metabolism (Zhang et al., 2022). The Vitamin D Receptor (VDR) activates CYP3A4/5 transcription through heterodimer formation, influencing cellular functions. (Qin & Wang, 2019). Furthermore, Estrogen Receptor alpha (ERα) regulates CYP3A4/5 transcription directly, playing a vital role in liver drug metabolism beyond reproductive functions. ESR1 forms regulatory networks with other transcription factors, contributing to drug metabolism modulation (Wang et al., 2019).

Post-transcriptional regulation

Post-transcriptional regulation significantly influences CYP3A4/5 activity, contributing to inter-individual variability in drug metabolism. Specific miRNAs can bind CYP3A4/5 mRNA, leading to mRNA degradation through RNA-induced gene silencing or translational repression by preventing ribosome binding or altering mRNA conformation (Correia de Sousa et al., 2019). Post-translational modifications, such as phosphorylation and ubiquitination, regulate CYP3A4/5 stability, activity, and subcellular localization. Phosphorylation by kinases can enhance or inhibit CYP3A4/5 activity, while ubiquitination targets these enzymes for proteasomal degradation (Frédérick & Simard, 2022). Protein degradation pathways, including the ubiquitin-proteasome system and autophagy, maintain cellular protein homeostasis and regulate CYP3A4/5 levels (Ohya et al., 2023). The ubiquitin-proteasome system is responsible for the majority of intracellular protein degradation. It involves the covalent attachment of ubiquitin molecules to target proteins, followed by their recognition and degradation by the 26S proteasome, which is essential for adjusting metabolic activity in response to various stimuli for CYP3A4/5 (Johnson, Su & Zhang, 2021). On the other hand, autophagy is a lysosome-dependent pathway that degrades long-lived proteins and entire organelles (Fleming et al., 2022). While it is generally considered a bulk degradation process, recent evidence suggests that autophagy can selectively target specific proteins, including CYP enzymes. In the context of inflammatory diseases, autophagy may contribute to the basal turnover of these enzymes and potentially mediate the clearance of aggregated or misfolded protein (Morgan et al., 2020). Cytochrome b5, an interacting protein, forms complexes with CYP3A4/5, influencing their catalytic activity and electron transfer (Bart & Scott, 2017; Yantsevich, Gilep & Usanov, 2008). Cytochrome b5 can also alter CYP3A4/5 metabolic pathways, affecting drug metabolite profiles and efficacy.

Regulation of enzyme activity and influencing factors

The intracellular activity of CYP3A4 and CYP3A5, critical drug-metabolizing enzymes, is intricately regulated and influenced by a myriad of factors, including drug interactions, hormone levels, genetic factors, dietary and environmental elements, disease states, and age and gender (Fernandez-Teruel et al., 2024; Peruzzi et al., 2024). Notably, drug interactions significantly impact CYP3A4/5 activity, with certain drugs capable of acting as inducers or inhibitors, directly influencing other drugs’ metabolic rates. Moreover, hormonal changes, particularly elevated cortisol levels during pregnancy, have been shown to directly affect CYP3A4/5 enzyme activity, leading to an increased rate of drug metabolism (Mlugu et al., 2022), which underscores the importance of considering hormonal effects on CYP3A enzyme activity during pregnancy, which can guide drug dose adjustments. Genetic factors, such as single nucleotide polymorphisms, also play a pivotal role in CYP3A4/5 expression levels and activity. For instance, CYP3A4*4 (A13871G) and CYP3A4*1B (A392G) have been found to decrease enzyme activity, resulting in slower metabolism. Wang et al. (2005) reported that CYP3A4*4 mutants reduce enzyme activity and drug metabolism, enhancing the efficacy of simvastatin. At the same time, Rodríguez-Antona et al. (2005) found that CYP3A4*1B mutants decrease the enzyme’s transcription rate, reducing its activity and increasing quinine metabolism. Dietary components, including lycopene, a naturally occurring carotenoid, and environmental chemicals can also significantly affect CYP3A4/5 enzymatic activity. Studies have demonstrated that lycopene can increase the metabolic activities of CYP2B, CYP2D, and CYP3A, rat homologous enzymes to CYP3A4/5, particularly at high doses (Nosková et al., 2020). Furthermore, disease states, such as diabetes, can influence CYP3A4/5 activity in the liver due to accompanying insulin resistance and metabolic disorders (Carr, Turner & Pirmohamed, 2021; Grandl et al., 2019; Lamba et al., 2002; Neuvonen, Niemi & Backman, 2006). Research using a diabetic rat model has shown that CYP3A4 activity decreases in the short term and increases in the long term (Wang et al., 2023), highlighting the complex effects of diabetes on CYP450 enzyme system activity and the need for precise medication use in diabetic patients. Lastly, age and gender are physiological factors that may alter CYP3A4/5 activity. Changes in drug metabolism rates in children and older people may result from the maturity or functional decline of the metabolic enzyme system (Hunt, Westerkam & Stave, 1992). Research indicates that aging male rats exhibit decreased activity of major CYP isoforms (Konstandi & Johnson, 2023), reflected in lower metabolism and higher drug substrate levels in the blood, emphasizing age as a crucial factor in CYP gene regulation. Additionally, gender differences, with women potentially having higher CYP3A4/5 activity due to elevated estrogen levels, may result in more rapid drug metabolism compared to men (Khatri et al., 2021). These multifaceted factors necessitate a comprehensive understanding and personalized approach to drug therapy, considering the complex interplay of influences on CYP3A4/5 activity.

Metabolic substrate drugs of drug interactions

CYP3A4/5, as crucial intracellular metabolic enzymes, are responsible for metabolizing numerous drugs and endogenous compounds (Kondža, Brizić & Jokić, 2024; Rodriguez-Antona et al., 2022). Metabolic substrate drugs refer to medications that require metabolism via CYP3A4/5, and their metabolic processes influence the pharmacodynamics and pharmacokinetics of these drugs in the body (Bolleddula et al., 2022; Loos, Beijnen & Schinkel, 2023). Drug-drug interactions (DDIs) denote the effects that may arise when two or more drugs are used together, including enhancement or attenuation of each other’s pharmacological effects and increased risk of adverse reactions (Bolleddula et al., 2022; Loos, Beijnen & Schinkel, 2023). DDIs occur when one drug interacts with another or multiple drugs in the body, leading to alterations in their pharmacokinetic parameters and consequent changes in pharmacological effects and toxicity (Kulkarni & Singh, 2024). Such interactions may amplify or diminish the efficacy of drugs, resulting in adverse drug reactions (ADRs) or even fatalities (Simal et al., 2024). Concurrent use of multiple medications by patients is not uncommon, and combination therapy has demonstrated significant efficacy in combating cancer, autoimmune diseases, and other prevalent conditions (Chandra et al., 2023; Francini et al., 2024; Karnam et al., 2024). Studies estimate that 6.7% of hospitalized patients in the United States experience severe adverse drug reactions annually, with a mortality rate of 0.32%, amounting to over 2,216,000 severe ADR cases among hospitalized patients. The incidence rate of drug metabolism interactions affecting drug efficacy accounts for approximately 40% of pharmacokinetic interactions (Lazarou, Pomeranz & Corey, 1998). Therefore, research on metabolic substrate drugs of CYP3A4/5 and their interactions with other drugs is of paramount importance for personalized and precision medicine in clinical drug therapy.

Substrate categories of CYP3A4/5 metabolism

As indispensable members of the cytochrome P450 enzyme family, CYP3A4/5 are integral to the metabolism of a vast array of substrates, which can be categorized as follows: Steroid hormones, such as cortisol (Van Keulen et al., 2020), testosterone (Wu et al., 2022), and estradiol (Cheng & Zhou, 2000). Additionally, numerous lipophilic drugs necessitate CYP3A4/5 for metabolism, including benzodiazepines (Rodriguez-Antona et al., 2022), calcium channel blockers (Fuhr et al., 2022), antibiotics (Lenard et al., 2023), and specific antiepileptic medications (Zhao et al., 2021a), all of which are vital in the treatment of a myriad of diseases. While CYP3A4 and CYP3A5 share a broad substrate specificity, there are notable differences in their affinity and capacity to metabolize certain drugs. For instance, CYP3A4 is the primary enzyme involved in the metabolism of the immunosuppressant cyclosporine, which is crucial in post-organ transplant therapy (Zhai et al., 2024). In the metabolism of the immunosuppressant tacrolimus, where CYP3A5 is more efficient than CYP3A4 (Hannachi et al., 2024). Furthermore, the metabolism of drugs like nifedipine and erythromycin is primarily attributed to CYP3A4 due to its higher expression levels in the liver and intestine (Khatri et al., 2021; Szabó et al., 2024). In conclusion, the substrate categories metabolized by CYP3A4/5 encompass a broad spectrum of drugs and bioactive molecules with extensive clinical applications, significantly influencing individual drug metabolism and the potential for drug interactions.

Mechanisms of drug interactions

Since CYP3A4/5 is involved in the metabolism of a multitude of drugs, they play crucial roles in mediating drug interactions. The primary mechanisms of these interactions include competitive and non-competitive inhibition, as well as induction (Salvatorelli, Minotti & Menna, 2023; Lin et al., 2001). Competitive inhibition, a key mechanism at the microscopic level, occurs when a drug binds to CYP3A4/5 and competes for the same binding site as substrates, which are the drugs requiring metabolism (Wright, Chenge & Chen, 2019). This competition prevents substrates from binding to the enzyme, reducing their metabolic rate. Warfarin is metabolized by CYP3A4/5, when co-administered with another CYP3A4/5 substrate fluconazole, the metabolism of warfarin is significantly inhibited due to competition for enzyme active sites. The binding of inhibitors to the enzyme can be either reversible or irreversible, a characteristic that depends on the drug’s specific properties and binding mode (Ozawa, 2002). In addition to competitive inhibition, non-competitive inhibition also significantly impacts CYP3A4/5 activity. Tofacitinib is metabolized mainly by CYP3A4 and myricetin has a strong non-competitive inhibition on the metabolism of tofacitinib in CYP3A4 (Ye et al., 2024). Here, inhibitors bind to alternative sites on the enzyme, causing conformational changes or alterations in the enzyme’s active site structure, diminishing the enzyme’s catalytic efficiency (Hlavica, 2023). It is notable that the impact of the inhibitor on the enzyme also can be reversible or irreversible, depending on the specific properties of the drug and its binding mode (Fontana, Dansette & Poli, 2005; Varghese et al., 2015). Reversible inhibition of CYP3A4/5 typically involves non-covalent interactions such as hydrogen bonds or electrostatic interactions between the enzyme and the inhibitor (Scaglione, 2013). This type of inhibition is characterized by the ability of the enzyme to regain its activity once the inhibitor concentration changes. For example, midazolam, a substrate for CYP3A4/5, can be competitively inhibited by natural product goldenseal, which binds to the same active site and competes for enzyme binding, reducing the metabolic rate of midazolam. However, upon withdrawal of goldenseal, the enzyme activity can be restored, demonstrating the reversible nature of this inhibition (Nguyen et al., 2023). In contrast, irreversible inhibition results from covalent modification of the enzyme, leading to permanent inactivation. This can occur through various mechanisms, such as the formation of a covalent bond between the inhibitor and the enzyme’s active site, which cannot be reversed even after the inhibitor is removed. Understanding the reversibility of these inhibition mechanisms is crucial for predicting drug-drug interactions and their potential clinical implications. For instance, time-dependent inhibition (TDI) such as the famous antihypertensive drug mibefradil, which is a form of irreversible inhibition, can lead to more severe drug interactions as the enzyme activity remains suppressed even after the inhibitor is removed (Rougée et al., 2023). This is particularly relevant for drugs that are long-term medications or when drugs with TDI properties are co-administered. Clinical CYP3A4/5 Mediated Drug Interactions (Hohmann et al., 2024). At a microscopic level, these inducers may interact with transcription factors within the cell nucleus, initiating or enhancing the transcription of CYP3A4/5 genes. This process results in increased expression of the enzymes, which subsequently leads to elevated substrate metabolism rates. Moreover, drugs can also directly interact with CYP3A4/5 proteins, affecting their activity. Such interactions can induce conformational changes in the enzyme or obstruct substrate binding, influencing the overall metabolism of the substrate (Denisov et al., 2021). Collectively, these mechanisms highlight the multifaceted nature of drug interactions involving CYP3A4/5 and underscore the importance of understanding these processes to optimize drug therapy and minimize adverse effects.

Clinical CYP3A4/5 mediated drug interactions

Ketoconazole is a broad-spectrum antifungal agent known to inhibit CYP3A4 activity (Wen et al., 2024). Consequently, when patients are co-administered ketoconazole and pioglitazone, a medication used for diabetes management, the metabolic rate of pioglitazone is slowed down by ketoconazole (Melillo et al., 2023). This interaction increases pioglitazone’s blood concentrations and a heightened risk of adverse reactions (Wang et al., 2022). Similarly, red yeast rice, a popular herbal remedy for cholesterol reduction, can inhibit CYP3A4 activity. This inhibition is primarily attributed to the component monacolin K, which is structurally analogous to lovastatin. This inhibition impacts the metabolism of statin drugs, such as atorvastatin and simvastatin, leading to elevated blood concentrations of these statins and a potential risk of adverse effects, including myopathy (Raguzzini et al., 2021). Another example of a significant drug interaction involving CYP3A4 is the co-administration of rifampicin, an antibiotic medication, and warfarin, an anticoagulant. Rifampicin induces CYP3A4 activity, which, when combined with warfarin, accelerates the metabolism of the anticoagulant (Fahmi, Abdelsamad & Elewa, 2016). This induction effect reduces warfarin’s blood concentration, potentially diminishing its efficacy and increasing the risk of thrombosis (Mar et al., 2022). In addition, tacrolimus C0/D (concentration/dose) increased 1.8-fold in CYP3A5 expressing individuals (CYP3A5*1/*1 and *1/*3) when used in combination with diltiazem, while it increased 1.3-fold in CYP3A5 non expressing individuals (CYP3A5*3/*3), These individual differences in tacrolimus metabolism, distribution, and responsiveness affect therapeutic efficacy (Yang et al., 2024). These examples illustrate the critical role of CYP3A4 and CYP3A5 in drug metabolism. Clinicians should adjust drug dosages based on a patient’s genotype particularly for drugs with narrow therapeutic indices, select alternative medications with fewer interactions, and implement strategies to monitor patients more closely for adverse effects during concurrent therapy in practical applications. Management strategies could involve the use of clinical decision support systems that incorporate pharmacogenomic data to predict and mitigate potential drug-drug interactions. By integrating these practical applications and management strategies, clinicians can improve drug efficacy and ensure patient medication safety.

CYP3A4 and CYP3A5 genetic polymorphisms and abnormal drug metabolism

CYP3A4 can form oligomers in the cell, but its basic functional unit is a single polypeptide with a large active site capable of binding multiple substrates simultaneously, contributing to its broad substrate specificity. This active site is flexible and can accommodate a variety of substrates, which is a key feature of CYP3A4’s role in drug metabolism (Johnson et al., 2014). Continuous research on the enzymatic activity of CYP3A4 has revealed that alterations in the relevant encoding genes significantly affect its metabolic capacity for drugs, demonstrating individual differences in drug response (Mohamed et al., 2024; Wm Te Loo et al., 2023). Exploring the enzymatic kinetics of different CYP3A4 substrates typically involves using in vitro experimental methods (Zientek & Youdim, 2015). Common approaches for in vitro preparation of CYP3A4 protein or cell systems include PCR and the construction of in vitro incubation systems, typically comprising recombinant CYP3A4 cells or microsomes, substrates, and cofactors (Chi et al., 2024). Techniques such as standard deviation centrifugation, insect cell transfection, calcium precipitation, and PBS centrifugation and resuspension are often employed to obtain target microsomal proteins, with protein immunoblotting or BCA assays used to assess CYP3A4 expression within cell microsomes (Johnson, Tanner & Tucker, 2000; Wang & Lan, 2022; Yang et al., 2018). Meanwhile, in vivo preparations often involve CYP3A4 gene insertion and transmission, followed by expression and measurement of protein expression on suitable vectors, such as adenoviral vector methods or humanized mouse models. Enzymatic kinetic parameters are critical indicators used to describe the properties of enzymatic catalysis. The maximum catalytic rate (Vmax) refers to the maximum rate of enzyme catalysis when the enzyme-substrate complex is saturated with substrate. The Michaelis constant (Km) represents the affinity between substrate and enzyme at half the maximum catalytic rate, where the enzyme catalysis rate is half of Vmax when the substrate concentration reaches Km. These parameters collectively depict the efficiency of the enzyme and the substrate’s affinity. Clearance rate (CL) is a parameter in the field of drug metabolism representing the rate at which substrates are cleared from the body per unit of time, calculated using Vmax and Km. Substituting depletion methods are commonly employed to determine the enzymatic kinetic parameters of different drugs in microsomal systems. These involve plotting graphs of the natural logarithm of the percentage of the residual drug against time to determine the first-order depletion rate constant (Kdep) for each concentration. By plotting the relationship between Kdep and substrate concentration, the Michaelis–Menten constant Km can be estimated. The maximum velocity (V max) is calculated using CL int × Km.

Effect of CYP3A4 genetic polymorphisms on drug metabolism

Through in-depth research on CYP3A4 enzymatic activity, alterations in the relevant encoding genes have been found to affect its metabolic capacity for drugs significantly, demonstrating individual differences in drug response. Against this backdrop, a joint research team from the First Affiliated Hospital of Wenzhou Medical University and Wenzhou Medical University explored the impact of CYP3A4 gene polymorphism on the effects of various drugs using methods such as gene screening, gene recombinant expression, and enzymatic kinetic studies, providing a reference for clinical medication.

Chemotherapeutic drugs

Acalabrutinib’s metabolism exhibits significant variability with CYP3A4 variants, such as *3, *11, *29, and *33, as shown in Fig. 2, showing 166% to 599.86% higher activity than the wild type, while *9, *14, *16, *19, *23, *24, *28, and *32 have 9.32–23.91% lower activity (Han et al., 2021a). Osimertinib metabolism is elevated in variants *29, *32, and *33 by 223.43% to 284.52%, contrasting with *2, *7, *8, *12, *13, *14, *15, *16, *17, and *18, which have 25.68% to 48.25% less activity (Gao et al., 2022). Cabozantinib’s metabolism is notably reduced in *6, *20, and *30, with the other 24 variants displaying considerable shifts (Han et al., 2021b). Midostaurin metabolism is increased in *4, *5, *10, *15, *16, *19, *23, *28, *29, *31, *32, and *33 by 101.61% to 253.31%, and decreased in others like *2, *14, *18, and *34 (Xu et al., 2023). Regorafenib metabolism varies, with *20 inactive and *5, *16, *19, *24, and *29 similar to wild type, and the other 20 variants showing significant differences. Ibrutinib is metabolized by variants *28 and *29 at 39–46% of wild type, with *3, *4, *9, *19, and *34 at 117.49% to 291.30% higher activity. Artemether-lumefantrine metabolism is mostly decreased by variants, with artemether clearance reduced to 6.7% to 66.1% and lumefantrine increased to 1.73 to 3.96 times wild type (Zhang et al., 2023). Quinine metabolism is diminished in *8, *12, *13, *17, *20, and *21 to less than 10%, while *15 and *29 have 131.1% and 316.5% higher activity. Vandetanib metabolism is absent in *6 and *30, with *4 at 95.81% of wild type and other variants showing increased or decreased activity.

Figure 2 Relative clearance of CYP3A4 variants in Chemotherapeutic Drugs.

Acalabrutinib, Osimertinib, Cabozantinib, Midostaurin, Regorafenib, Ibrutinib, Artemether-lumefantrine, Quinine, Vandetanib.

Endocrine therapy drugs

Azelastine, an antihistamine for allergies, is metabolized by CYP3A4 to its active metabolite, with variants *3, *11, *18, *23 showing 1.98–3.85 times higher activity and *17 showing 17.36% lower activity shown in Fig. 3. Marcitentan, used for pulmonary hypertension, is also metabolized by CYP3A4, with variants *2, *3, *4, *5, *8, *10, *15, *16 having 102–504% higher activity, *7, *14, *18, *28–*32 having 50%-91% lower activity, and *9, *11-*13, *17, *20–*24, *33, *34 showing 5–45% lower activity (Li et al., 2019). Saxagliptin, an anti-diabetic medication, is metabolized by CYP3A4 to 5-hydroxy saxagliptin, with variants *11, *12, *17, *20, *23 having 1.91–7.90% lower clearance, *2 showing 10% lower 5-hydroxy saxagliptin formation, and *3, *4, *7, *8, *9, *13, *16, *18, *24, *29, *31, *32, *34 having 24.16–45.38% lower clearance, and *5, *10, *14, *19, *28, *33 classified as a mild reduction group (Liu et al., 2022).

Figure 3 Relative clearance of CYP3A4 variants in Endocrine therapy drugs.

Azelastine, Marcitentan, Saxagliptin.

Central therapeutic drugs

Lidocaine, an antiarrhythmic for ventricular arrhythmias and local anesthesia, is metabolized by CYP3A4 to MEGX, as shown in Fig. 4, with variants *11, *14, *15, *18, *19, *23, *29, *31, *32, *34 showing 123.27–213.61% higher intrinsic clearance, and *2, *5, *9, *16, *24 having 27.93–67.93% lower clearance (Fang et al., 2017). Istradefylline, an A2A receptor antagonist for Parkinson’s, is metabolized by CYP3A4, with *29 showing increased activity, *2, *3, *14, *31, *32 having 50–90% of wild-type activity, and *33, *34 having 35–38%, while *17 is nearly inactive (Hu et al., 2022). Brexpiprazole, an antipsychotic, is metabolized by CYP3A4 to DM-3411, with *2, *3, *7, *8, *9, *10, *11, *12, *13, *16 having 16.54–59.06% lower clearance than the wild type, *17 and *20 nearly inactive, and *14, *15 showing slightly higher, non-significant clearance (Chen et al., 2020). These CYP3A4 polymorphisms can significantly impact drug metabolism, efficacy, and safety, emphasizing the importance of personalized medicine.

Figure 4 Relative clearance of CYP3A4 variants in Central therapeutic drugs.

Istradefylline, Brexpiprazole, Lidocaine.

Circulatory system drugs

Sildenafil, a PDE5 inhibitor for pulmonary arterial hypertension and erectile dysfunction, is metabolized by CYP3A4 to N-desmethyl sildenafil. As shown in Fig. 5, variants *3, *10, *14, *15, *19, *32, *33 have 110–140% higher metabolic activity, while *24 has significantly reduced activity, nearly inactive, and *2, *5 have about 50% decreased activity (Tang et al., 2020). Amiodarone, a class III antiarrhythmic, is primarily metabolized by CYP3A4 to desethylamiodarone. Variants *17 and *24 have significantly reduced activity (11.07% and 2.67% of wild type), while *34, *15, *31, *29, *32, *18, *14, *10, *23, *19, *2, and *11 have increased activity (155.20~435.96% of wild type), with *2 and *11 showing a 289.48% and 435.96% increase, respectively. The remaining seven alleles, *4, *28, *5, *33, *3, *9, and *16, have similar activity to wild type (Yang et al., 2019).

Figure 5 Relative clearance of CYP3A4 variants in Circulatory system drugs.

Sildenafil, Amiodarone.

Effect of CYP3A5 genetic polymorphisms on drug metabolism

CYP3A5 shares 84.1% amino acid homology with CYP3A4, but its expression level in the liver is much lower than that of CYP3A4. To date, 34 variants of CYP3A5 have been identified, of which CYP3A5*6, which arises from a splice variant of 14690G > A, is a nonfunctional protein and is predominantly found in the African American population (Ranasinghe et al., 2024), and CYP3A5*8 and CYP3A5*9 are defective coding alleles that result in a reduction of CYP3A5 activity. CYP3A5*3 is the most studied and common variant of the CYP3A5 gene, which is predominantly found in European populations, and CYP3A5*3 may result in loss of CYP3A5 functional activity (*3/*3), partial deletion (*1/*3) or normal expression (*1/*1). The following pharmacokinetic differences in drugs affected by polymorphisms in the CYP3A5 gene will be explored below.

Atorvastatin is a 3-hydroxy-3-methylglutaryl coenzyme A reductase inhibitor, which can stabilize atherosclerotic plaques and lower lipid indices by inhibiting HMG-CoA reductase and decreasing cholesterol synthesis in the liver and can lower LDL-C by increasing the number of LDL receptors on the surface of hepatocytes and accelerating cholesterol Atorvastatin is used for the treatment of patients with primary hypercholesterolemia, as it reduces triacylglycerol (TG) and raises HDL-C by increasing the number of LDL receptors on the surface of hepatocytes, and accelerates cholesterol catabolism. Atorvastatin is mainly hydroxylated by CYP3A4 and CYP3A5 as well as CYP2C8 to produce the two main active metabolites, 2-hydroxy (2-OH) atorvastatin and 4-hydroxy atorvastatin. It was shown that the CYP3A5*3 genotype significantly affects the pharmacokinetics of both atorvastatin and 2-OH atorvastatin, and this was confirmed by comparing the CYP3A5*3/*3, CYP3A5*3, and CYP3A5*3 genotypes. *3, CYP3A5*1/*1, and CYP3A5*1/*3 carriers, it was found that the CYP3A5*3/*3 mutase enzyme activity was reduced compared to CYP3A5*1/*1 and that carriers of CYP3A5*3/*3 exhibited more than two-fold higher Cmax and AUCIN of atorvastatin than the wild type (Park et al., 2022).

Fentanyl is an opioid receptor agonist commonly used for all kinds of pain and surgery and other postoperative and surgical procedures of analgesia. It also can be used as an anesthesia adjuvant, but too large of a dose can cause addiction; in May 2019, the Chinese government formally merged fentanyl into the whole class of listed substances.

Fentanyl is metabolized to desmethyl fentanyl and hydroxydesmethylfentanyl primarily through CYP3A4 and CYP3A5.2022. Williams et al. (2022) investigated the effect of CYP3A5 gene polymorphisms on the pharmacokinetics of fentanyl use in children, and found that clearance was not associated with the CYP3A4*1G gene variant but was not associated with CYP3A5*3 or CYP3A5*6 allele variants. There was evidence of loss of physiological function effects for both CYP3A5*3 and CYP3A5*6 variants, and the data indicated that the clearance of fentanyl was lower in carriers of CYP3A5*3/*3, CYP3A5*3/*6, and CYP3A5*6/*6 compared to carriers of CYP3A5*1/*3 and CYP3A5*1/*6, so the study demonstrated that fentanyl was more effective in children than in those with CYP3A5*1/*3 and CYP3A5*1/*6. This study demonstrates that CYP3A5 nonfunctional variants affect fentanyl clearance in the pediatric population (Williams et al., 2022).

Vincristine is an alkaloid extracted from periwinkle that targets microtubules and stops tumor cells from dividing by preventing spindle microtubule formation through inhibition of polymerization of microtubule proteins. It is mainly used for the treatment of childhood lymphoblastic leukemia and non-Hodgkin’s lymphoma. Vincristine metabolism is linked to CYP3A5, and in 2018, Skiles et al. (2018) found that the majority of Kenyan children with a high CYP3A5-expressing genotype (*1/*1) had 58% less exposure to vincristine than Kenyan children with a low-expressing genotype (*3, 6, or 7/*3, 6, or 7), which may lead to the development of child drug resistance, and carriers of the CYP3A5 high-expressing genotype had a significantly reduced chance of developing vincristine-induced peripheral neuropathy (VIPN); therefore, genetic variation in CYP3A5 affects the metabolism of vincristine and susceptibility to toxicity.

Apatinib is a VEGFR2 inhibitor that prevents the generation of new blood vessels in tumor tissues by inhibiting tyrosine kinase activity and is the world’s first orally available small molecule targeted drug that can be safely and effectively used in the treatment of advanced gastric cancer, as well as for the treatment of solid tumors, such as breast, gynecological and lung cancers. Apatinib is metabolized to inactive E-3-hydroxyapatite primarily through CYP3A4 and CYP3A5. The disposition process of apatinib in vivo has large individual differences Yang et al. (2023) established a pharmakinetic (PK) model of adult cancer patients treated with apatinib, which was found that the apparent clearance and apparent volume of distribution of CYP3A5-expression (CYP3A5*1/*1 and CYP3A5*1/*3) were lower than that of CYP3A5-non-expression (CYP3A5*3/*3), while the dose should be 33.33–50.00% lower than in CYP3A5 non-expression (Yang et al., 2023). Therefore, the latter would require a higher dose of the drug to achieve a therapeutic effect during treatment.

Sorafenib is a multi-targeted tyrosine kinase inhibitor that exerts antitumor effects by inhibiting tumor angiogenesis and cell proliferation and is used in the treatment of patients with advanced hepatocellular carcinoma who are not suitable for surgery. CYP3A4 and CYP3A5 mainly metabolize sorafenib in the human body. Its main metabolite exists in the form of pyridine N-oxide and a small number of people may experience adverse reactions such as hypertension, edema, and hand-foot syndrome. Individuals with the CYP3A5*3 variant are less able to metabolize sorafenib compared to the wild type (Guo et al., 2018). Song et al. (2022) found that the enzyme catalytic activities of the CYP3A5*2, CYP3A5*3A, CYP3A5*3C, CYP3A5*4, CYP3A5*5, and CYP3A5*7 variants were all lower than those of the wild-type so that the intrinsic clearance of sorafenib was reduced, while the CYP3A5*6 allele exhibited a higher enzyme catalytic activity than the wild-type. Alleles with high catalytic activity help to inhibit the proliferation of tumor cells and induce their apoptosis (Song et al., 2022).

In addition, CYP3A5 polymorphisms help predict disease susceptibility. Some studies have reported that CYP3A5 polymorphisms are closely associated with prostate cancer, and 3A5 gene polymorphisms can be used as biomarkers to help prevent and treat prostate cancer (Zhenhua et al., 2005). For instance, a meta-analysis including six case-control studies with 2,522 cancer patients and 2,444 healthy controls showed that CYP3A5*3 polymorphisms were significantly associated with an increased risk of prostate cancer under two genetic models (GG + AG vs. AA: OR = 1.53, 95% CI [1.23–1.90], P = 0.000; GG vs. AA: OR = 1.46, 95% CI [1.14–1.87], P = 0.000). This suggests that genetic variations within the CYP3A5 gene can substantially affect an individual’s susceptibility to prostate cancer, emphasizing the importance of considering these genetic factors in clinical practices (Liang et al., 2018). Another study has shown that CYP3A5 regulates prostate cancer cell growth by facilitating nuclear translocation of AR, highlighting the enzyme’s direct role in cancer progression (Mitra & Goodman, 2015). Therefore, when administering drugs in the clinic, appropriate drug therapy should be selected for the mutation of the CYP3A5 allele to achieve individualization of drug dosage, thus improving the efficiency of drug utilization and reducing drug toxicity.

Effect of individual differences on drug metabolism

During drug therapy, individual differences between patients, particularly with drugs that have a narrow therapeutic window, can significantly impact drug plasma concentrations, potentially leading to adverse drug reactions or therapeutic failure. These individual differences mainly encompass gender, age, race, and disease status (Eichelbaum, Ingelman-Sundberg & Evans, 2006). Gender is increasingly recognized as a determinant of pharmacokinetic (PK) and pharmacodynamic (PD) profiles. Reports indicate that females metabolize drugs faster than males, partly due to gender differences in metabolic enzyme activity, which includes a higher level of CYP3A4 protein expression in the liver of females (Yoon et al., 2021). Given that CYP3A4 and CYP3A5 account for over half of the total CYP content in the human liver and metabolize more than half of prescribed drugs (Urban et al., 2018), it is not surprising that clearance of most drugs tends to be higher in women than in men. Additionally, gender differences in renal function, often influenced by body weight, result in a higher glomerular filtration rate in males than in females (Layton & Sullivan, 2019). Age also exerts a crucial influence on drug metabolism. Hepatic levels of total CYP enzymes begin to decline around 40 years old, with a decrease in CYP3A content observed in the liver with increasing age. While substrate metabolism rates for CYP1A2 and CYP2C19 decrease with age, those for CYP2A6, CYP3A4, CYP2C9, and CYP2D6 do not show significant changes (Corton et al., 2022). Furthermore, older age is associated with a reduction in hepatic blood flow by approximately 35% and a decrease in liver size by 24–35% (Wynne, 2005). Consequently, the clearance of liver-metabolized drugs declines with the reduced liver size. Additionally, age-related declines are observed in intestinal absorption, glomerular filtration, tubular secretion, passive reabsorption, renal elimination, and hormonal levels, including gonadal hormones, thyroid hormones, insulin, glucagon, and glucocorticoids. These hormonal changes can affect CYP expression and the functionality of the hormonal system. Racial differences can result in varying capacities for drug metabolism. For instance, midazolam area under the curve (AUC) levels were observed to be 27% higher in Caucasian populations compared to South Asian populations when administered at the same dose, with clearance of midazolam being significantly higher in South Asians than Caucasians (Van Dyk et al., 2018) Genetic variations between racial groups also contribute to differences in metabolic activity. For instance, African Americans exhibit the lowest metabolic activity for warfarin among Caucasian, Asian, and African American populations (Ohara et al., 2019). Moreover, disparities in living environments and dietary habits between races can influence drug metabolism. Disease states frequently impair the ability to metabolize drugs. For example, patients with hypoalbuminemia who utilize drugs with high protein binding exhibit elevated free drug concentrations in comparison to healthy individuals. In conditions such as renal insufficiency and cirrhosis, the capacity to metabolize drugs is diminished.

In conclusion, individual differences, influenced by a multitude of factors, including gender, age, race, and disease status, play a pivotal role in drug metabolism. Consequently, it is of the utmost importance for healthcare professionals to individualize patient treatment in clinical settings, tailor drug administration protocols, and monitor blood concentrations promptly to ensure the efficacy and safety of the drugs administered.

CYP3A4/5 in individualized drug therapy

In personalized drug therapy, CYP3A4 and CYP3A5 are key enzymes in drug metabolism, and their genetic polymorphisms have a significant impact on drug response and safety. The clinical significance of CYP3A5 genetic variants is exemplified by the differential dosing requirements of immunosuppressive drugs like tacrolimus in post-transplant patients. Individuals with CYP3A5 expressor genotypes (CYP3A5*1/*1 and *1/*3) metabolize some CYP3A substrates more rapidly than CYP3A5 nonexpressors (e.g., *3/*3). For instance, a study demonstrated that the dose of tacrolimus required for CYP3A5 non expressing individuals with normal CYP3A4 expression levels is approximately twice that of CYP3A5 expressing individuals, while low CYP3A4 expressing individuals require approximately four times the dose (Zong et al., 2024). The Clinical Pharmacogenetics Implementation Consortium (CPIC) has published guidelines recommending lower initial tacrolimus doses for CYP3A5*3/*3 patients to avoid toxicity and achieve therapeutic efficacy (Birdwell et al., 2015). Understanding the genetic variations of these enzymes is crucial for customizing drug doses, reducing adverse reactions, and improving efficacy. This section will explore how to personalize drug therapy through genetic testing and the role of CYP3A4/5 in guiding clinical drug use.

Drug metabolism genetic testing and personalized medication

Cytochromes P450 3A4 and 3A5 are involved in the metabolism of many drugs, and there are many genetic variants of these two enzymes due to, for example, nucleotide mutations. Currently, commonly used methods for detecting polymorphisms in these two genes include DNA sequencing analysis, Polymerase chain reaction-restriction fragment length polymorphism (PCR-RFLP), capillary polymerase chain reaction (CPR-RFLP), and capillary polymerase chain reaction (CPCR). Phism, PCR-RFLP, Capillary Electrophoresis (CE), Amplification Refractory Mutation System PCR (ARMS-PCR), Real-time Reverse Transcriptase-Polymerase Chain Reaction (RT-qPCR) DNA sequencing is the process of chemically sequencing DNA.

DNA sequencing is the conversion of DNA chemical signals into digital signals that computers can process. DNA sequencing has been used for nearly 40 years. Currently, the most commonly used is the second-generation sequencing technology, also known as high-throughput sequencing technology, which is fast and can sequence millions to billions of DNA molecules at the same time in a single experiment through automation and parallelization; it is characterized by short read lengths, usually in the range of 50–600 bp; due to high throughput and automation, its cost per sequencing site is lower; however, its short read lengths may lead to certain regions of repetitive sequences or structural variants are difficult to analyze. Recently, third-generation sequencing (TGS) technologies have shown clear advantages in resolving highly repetitive or polymorphic regions in pharmacogenomics.

PCR-RFLP is a combination of PCR technology, RFLP analysis, and electrophoresis. PCR and the specific endonuclease digests first amplify the target DNA. It cuts the amplified product into fragments of different sizes, which are then separated by gel electrophoresis. Then, the sizes and positions of the various restriction fragments can be distinguished under ultraviolet light. This method is highly accurate and simple but requires manual operation and is not suitable for high-throughput and large-scale sample detection.

CE uses a high-voltage electric field as the driving force and a capillary tube as the separation channel and realizes separation based on the different migration rates of different protein molecules in the electric field. It has a wide range of applications and can be used for genetic diseases caused by gene mutations, oncogenes, and infectious diseases. It also has the advantages of rapid, high throughput, easy quantification, automation, and suitability for analysis of micro samples. At the same time, CE also has some limitations: it is more sensitive to the initial conditions of the problem, the selection of the initial population is more demanding, and due to the complexity of the operation mechanism of CE, the computational complexity of its algorithm is correspondingly higher. It can be seen that all three methods have the disadvantages of complicated operation, longer time-consuming, high cost, low detection sensitivity, and easy to cause false positives or false negatives (Cheng, 2016; Li & Li, 2014; Ribeiro, Martins & Grazina, 2017).

ARMS-PCR technology is based on allele-specific extension reaction, which can be performed only when the 3′ end base of an allele-specific primer is complementary to the base at the mutation site. It is a new method developed on the basis of PCR, which can detect various point mutations and known mutations in DNA. It has the advantages of simple operation, short time, high sensitivity, and low cost. However, ARMS-PCR has low throughput, cannot detect unknown mutations, and is not suitable for typing SNPs with too high or too low GC content near the site.

RT-qPCR is a technique that combines reverse transcription and real-time fluorescent quantitative PCR. The most common methods are fluorescent dye and probe methods, which emit a signal when binding to amplified DNA or RNA molecules, allowing quantification of the target sequence. There is a one-to-one correspondence between the probe and the template, so its probe method has the advantages of high specificity and sensitivity. However, the quenching is difficult to thorough, the background is high, and the detection results are difficult to judge the actual amplification characteristics (Zucha, Kubista & Valihrach, 2021). The fluorescent dye method has no selection of DNA templates and may produce false positive results.

CYP450, as a hemoglobin with many isozymes and isoforms, is responsible for the metabolism of more than two-thirds of exogenous substances. With advances in sequencing technology, the main ones identified so far are CYP1A1, CYP1A2, CYP2A6, CYP2B6, CYP2C9, CYP2C19, CYP2D6, CYP3A4, CYP3A5 and CYP3A7 (Naidoo, Chetty & Chetty, 2014). Single nucleotide polymorphisms (SNPs) are major variants on the genome that may lead to increased or decreased enzyme activity, resulting in different drug metabolizing abilities between individuals. The population is categorized into four groups based on the genes carrying different forms of polymorphisms: extensive metabolizers (with normal enzyme functional activity), poor metabolizers (with loss of enzyme functional activity), ultra-rapid metabolizers (with increased enzyme functional activity), and intermediate metabolizers (with decreased enzyme functional activity) (Zanger & Schwab, 2013). Each individual is unique, and therefore, drug class selection and drug dosage must be adapted to genomic diversity and other determinants of drug response in order to minimize toxic reactions and improve therapeutic efficacy.

The role of CYP3A5 in guiding clinical drug use

Genetic variation in the CYP3A5 gene can alter the pharmacokinetics and pharmacodynamics of drugs during clinical drug administration. Loss of CYP3A5 function results in decreased drug clearance and increased blood levels, while gain of CYP3A5 function results in increased drug clearance and decreased blood levels. There is a strong correlation between the expression of CYP3A5 and race, with the allele with the highest prevalence among Asian Koreans being CYP3A5*1/*3, and white patients are more likely to have a CYP3A5*3/*3 non-expressing genotype than black patients.

Tacrolimus is currently the only drug used in the clinic based on recommendations for CYP3A5 genotype guidance. Genetic polymorphisms in CYP3A5 have been reported to affect the immunosuppressive efficacy of tacrolimus in patients after liver transplantation by altering tacrolimus bioavailability and metabolism rate after liver transplantation (Birdwell et al., 2015). In 2015, the Clinical Pharmacogenetics Implementation Consortium (CPIC) published guidelines on the use of pharmacogenomic testing in tacrolimus administration. Tacrolimus is a macrolide immunosuppressive drug used primarily after solid organ and hematopoietic stem cell transplantation. The arrival of effective blood concentrations may be delayed after tacrolimus administration in CYP3A5-expressing populations compared to CYP3A5 non-expressing populations. Therefore, during clinical use of tacrolimus, in order to maintain tacrolimus blood levels at therapeutic levels, CYP3A5 non-expressors require a standard recommended starting dose, whereas CYP3A5-expressors require a higher recommended starting dose. In summary, CYP3A5 guides the clinical use of tacrolimus and makes drug use safer.

Conclusions

CYP3A4 and 3A5 are important drug metabolism enzymes in the human body, involved in the metabolism of approximately 30–50% of known drugs. The genetic polymorphism of CYP3A4/5 and drug interactions significantly impact drug metabolism and interactions. Research has found that various transcription factors and ligands, such as PXR, regulate the gene expression of CYP3A4 and CYP3A5. Ubiquitination and protein degradation pathways play important roles in regulating the activity and stability of CYP3A4 and CYP3A5. When comparing the structures of CYP3A4 and CYP3A5 proteins, a remarkable high degree of homology is observed. This structural similarity leads to the manifestation of comparable metabolic properties when these enzymes interact with the same substrates. In fact, the studies on the polymorphisms of CYP3A5 have shown a significant resemblance to those of CYP3A4 in many aspects. This similarity implies that the findings from CYP3A4 polymorphism research can offer valuable references for understanding CYP3A5 polymorphisms. It is important to note that the same variants of these enzymes can exhibit distinct enzymatic kinetic characteristics depending on the specific substrates involved. This emphasizes the fact that we should not restrict our investigations to probe substrates only. Each substrate may interact with the enzyme variants in a unique way, thereby influencing the enzymatic kinetic behavior. Therefore, the research on enzymatic kinetics is a complex and non-repetitive task that requires a comprehensive exploration of different substrates and their interactions with the enzyme variants. The commonly used detection methods currently include DNA sequencing, PCR-RFLP, capillary electrophoresis, ARMS-PCR, and RT qPCR. In addition, living environment, dietary habits, and disease status can also affect drug metabolism ability. Individual differences play an important role in drug metabolism. Therefore, personalized treatment is necessary for clinical use, as well as adjusting medication regimens and monitoring blood drug concentrations to ensure the efficacy and safety of drugs. By studying mutations and polymorphisms in the CYP3A4 and CYP3A5 genes, we found that these gene mutations can significantly affect drug metabolism ability and individual response to drugs. Different mutations and polymorphisms can lead to changes in enzyme activity, thereby affecting drug clearance and pharmacokinetic parameters. In addition, individual differences, racial differences, and disease status can also have an impact on drug metabolism ability. Therefore, in clinical medication, individualized treatment should be considered, and the medication regimen should be adjusted based on the patient’s genotype and drug metabolism to ensure the efficacy and safety of the drug. With the advancement of science and technology, the application of artificial intelligence and big data in the medical field is becoming more and more widespread. It plays a comprehensive analysis and auxiliary role in developing personalized drug treatment. Artificial intelligence can use different algorithms to combine different types of omics data (such as genomics, proteomics, microbiomics, metabolomics, and phenotype data) with other information to build predictive models, discover potential patterns, and dynamically adjust the patient’s medication dosage. It can also provide a more accurate auxiliary diagnosis for clinical decision-making by leveraging the patient’s physiological and peripheral environmental status, helping to develop personalized medication plans for each patient.

According to Pharmavar data, the CYP3A4/5 allele pool is still expanding. However, the corresponding functional studies still need to catch up, and many alleles are functionally unknown, thus failing to provide basic data for genotype-metabolism phenotyping. The main reason for this situation is that, on the one hand, the popularization of genome sequencing technology has provided a fast channel for high-throughput analysis of biological samples, and the data have accumulated rapidly. According to empirical analysis, the frequency of discovery of novel CYP3A4/5 alleles is about 1/4,500, and the popularity of sequencing provides more than “digestible” information for discovering new variants. On the other hand, the functional study of alleles involves cross-disciplines such as molecular biology, enzymology, and analytical chemistry, which is technically complicated to advance the related research. In particular, the characteristics of CYP3A4/5 itself determine its catalytic activity in the form of microsomes. This mixed enzyme reaction system brings difficulties in standardization. To mitigate this issue, researchers could utilize the high-throughput screening (HTS) techniques. HTS allows for the rapid assessment of a large number of alleles for their metabolic activity. The development of specific fluorescent probes and miniaturized 384-well formats has accelerated the ability to develop robust in vitro assays in HTS format, which can be used to evaluate the potential drug–drug interactions mediated by CYP enzymes, including CYP3A4 (Chen et al., 2024; Kariv, Fereshteh & Oldenburg, 2001). Furthermore, the rational engineering of CYP3A4 fluorogenic substrates enables real-time sensing and functional imaging of CYP3A4 activities in various biological systems, facilitating functional studies on certain CYP3A4/5 alleles (He et al., 2023). In addition to HTS, the development and utilization of more advanced in vivo models such as humanized mouse models could provide a more accurate representation of the complex metabolic processes involved. Replacing the mouse’s own CYP450 genes with human CYP3A4/5, can offer valuable insights into the functional differences between various alleles (Ince et al., 2013). These models can be used to study the metabolic clearance of drugs and the potential for drug-drug interactions in a more human-like system. Moreover, the functional characterization of CYP3A4 variants by assessing midazolam 1′-hydroxylation and testosterone 6β-hydroxylation provides guidance for improving drug administration protocols (Kumondai et al., 2021). Although there are already corresponding solutions available, as a result of the above two aspects, the increase of CYP alleles is greater than the rate of functional research. The precise application of CYP3A4/5 based drugs is still difficult due to the lack of a complete “genotype-function-metabolism-phenotype” data chain. Therefore, the integration of transcriptomics, proteomics, epigenetics and metabolomics in genome-wide association studies may help to study the allele function more efficiently and accurately and promote the maximization of therapeutic efficacy and minimization of toxic side effects of drugs to achieve safe, effective and low-cost individualized drug therapy and jointly promote human health.

The research boom in precision medicine has driven the development and application of various genetic testing technologies for individualized drug use. Real-time quantitative fluorescent PCR, gene chips, nucleic acid mass spectrometry, and other kits and methods have promoted the precise use of clinical drugs, including cardiovascular and cerebrovascular drugs, immunosuppressants, and psychotropic drugs. However, these kits or assays often target genotypes with high prevalence but do not include low-frequency variants and are difficult to implement on a large clinical scale due to testing costs. In addition, current drug development and use typically uses a single dose that is effective for most of the population, with simple dose adjustments for a small number of patients. This practice will continue for many drugs, especially those that are inexpensive and have a broad therapeutic index. Improved dose individualization should focus on expensive drugs with small therapeutic indexes and high used volumes, such as cancer immunotherapies, immunomodulators, and anticoagulants. Precision medicine has been practised for several years and has gradually evolved from early single-molecule markers to histology and biological fingerprinting. In recent years, the integration of artificial intelligence technology and medicine has even promoted the leapfrog development of precision drug applications, and machine learning can provide patients with precise drug program output. With the development of multi-omics research technology and in the context of high-throughput data, precision medicine will move toward smarter, more accurate and more efficient development.

We thank Institute of Molecular Toxicology and Pharmacology of Wenzhou Medical University for consultation availability that supported this work.

Additional Information and Declarations

Competing Interests

Author Contributions

Data Availability

The authors declare that they have no competing interests.

Yuqing Zhang conceived and designed the experiments, performed the experiments, analyzed the data, prepared figures and/or tables, authored or reviewed drafts of the article, and approved the final draft.

Ziying Wang performed the experiments, analyzed the data, prepared figures and/or tables, and approved the final draft.

Yuchao Wang performed the experiments, analyzed the data, prepared figures and/or tables, and approved the final draft.

Weikai Jin performed the experiments, analyzed the data, prepared figures and/or tables, and approved the final draft.

Zheyan Zhang performed the experiments, analyzed the data, prepared figures and/or tables, and approved the final draft.

Lehao Jin performed the experiments, analyzed the data, prepared figures and/or tables, and approved the final draft.

Jianchang Qian conceived and designed the experiments, authored or reviewed drafts of the article, and approved the final draft.

Long Zheng conceived and designed the experiments, authored or reviewed drafts of the article, and approved the final draft.

The following information was supplied regarding data availability:

This is a literature review.

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
