# Peer review of "CYP3A4 and CYP3A5: the crucial roles in clinical drug metabolism and the significant implications of genetic polymorphisms"

_PeerJ, doi:10.7717/peerj.18636_

## Round 0.1 · original submission · Major Revisions

Please address the issues pointed by the reviewers and amend your manuscript accordingly.

Reviewer 1 ·

Basic reporting

no comment.

Experimental design

no comment.

Validity of the findings

no comment.

Additional comments

no comment.

Annotated reviews are not available for download in order to protect the identity of reviewers who chose to remain anonymous.

·

Basic reporting

The abstract provides a good summary but could be clearer in explaining the practical implications of CYP3A4/5 polymorphisms in personalized medicine. Consider rephrasing the sentence, “Such differences can affect both the efficacy and safety of the drug” to something more specific, like “These polymorphisms can lead to either increased drug toxicity or reduced therapeutic effects, requiring dosage adjustments based on genetic profiles.”

The introduction provides a solid foundation, but some sentences are overly complex and may confuse readers. For example, the sentence “CYP3A4/5 are involved in in vivo drug metabolism through a variety of pathways…” could be broken down for clarity. Consider splitting the sentence to ensure that each metabolic pathway is introduced clearly.

The section discussing individual differences in drug metabolism based on age and gender is highly relevant but lacks direct references to recent studies. Strengthening this with citations would improve its authority.

Experimental design

The methodology section outlines the databases used but does not specify exclusion criteria in enough detail. For instance, the manuscript could benefit from adding a sentence like, “Studies that did not directly address CYP3A4/5’s role in drug metabolism or lacked clinical data were excluded from this review.” This will enhance the transparency and replicability of the methodology.

This section is informative but could provide more detail on the experimental procedures used in some of the referenced studies, particularly around transcriptional regulation. For example, it mentions PXR and HNF4α’s role in regulation, but it does not provide enough context for how these mechanisms were studied. A more detailed explanation of the methods used in these studies would improve the review’s depth.

Validity of the findings

Page 24, Lines 487-495 (Genetic Testing and Personalized Medicine): This section could emphasize more clearly how genetic testing directly influences clinical outcomes. For example, you could elaborate on how variants such as CYP3A53/*3 and CYP3A51/*1 require different drug dosages in post-transplant patients, giving a more specific example from clinical settings (e.g., tacrolimus dosing).

Page 27, Lines 600-615 (Limitations of Current Research): The section discussing the lack of functional studies on certain alleles is crucial, but it needs to be expanded. You could suggest potential methodologies that future studies might use to close this gap, such as high-throughput screening or more advanced in vivo models. For example, the phrase “this mixed enzyme reaction system brings difficulties in standardization” could be followed by a discussion of possible approaches to mitigate this issue.

Additional comments

While the manuscript covers a significant topic in pharmacogenomics, it requires major revisions. The main areas that need improvement are the clarity of the language, especially in complex sentences, and the depth of explanation regarding the methods and their clinical applications. The inclusion of specific examples from recent studies, more critical analysis of the findings, and clearer transitions between sections will make the manuscript stronger and more impactful.
Thus, I recommend a major revision due to the need for enhanced clarity, more detailed methodological explanations, and additional references to strengthen the claims made in the review.

Reviewer 3 ·

Basic reporting

See additional comments

Experimental design

See additional comments

Validity of the findings

See additional comments

Additional comments

In this review, Zheng and coworkers provide an extensive review of the roles of CYP3A4 and CYP3A5 enzymes in drug metabolism, with a particular emphasis on genetic polymorphisms. The article explains how these enzymes metabolize about 30-50% of drugs and how genetic variations affect their enzymatic activity, leading to inter-individual differences in drug efficacy and safety. The review covers detection methods for genetic variants, regulatory factors influencing enzyme activity, and the clinical applications of understanding these polymorphisms in personalized medicine. It also discusses drug-drug interactions, enzyme kinetics, and the importance of individualized treatment based on genetic testing. However, the review does not clearly establish what new insights or unique perspectives it contributes compared to existing literature on CYP3A4/5 polymorphisms. The article needs to make some revisions to address these issues and then could be considered for publication.

1. When discussing protein degradation pathways, it is important to differentiate between the ubiquitin-proteasome system and autophagy in terms of their effects on CYP3A4/5. Each system plays a distinct role in protein homeostasis. A concise explanation of their specific contributions to CYP3A4/5 regulation and turnover would be beneficial for a comprehensive understanding.
2. In the “Substrate Categories of CYP3A4/5 Metabolism” section, it is advisable to include specific examples that highlight the differences in substrate metabolism between CYP3A4 and CYP3A5. This will aid in clarifying their respective roles and functionalities.
3. For “Mechanisms of Drug Interactions” part, the description of non-competitive inhibition as inhibitors binding to non-active sites on the enzyme, leading to conformational changes that affect substrate binding and enzyme activity, is accurate. However, to enhance the comprehensiveness of the information, it is beneficial to clarify whether the impact of the inhibitor on the enzyme is fully reversible or if there are specific studies on enzyme-substrate-inhibitor interactions that provide additional insights into this mechanism.
4. When addressing reversible and irreversible inhibition, it is crucial to clearly differentiate between these types of inhibition in various contexts. Reversible inhibition typically involves hydrogen bonds or electrostatic interactions, allowing enzyme activity to be restored when inhibitor concentrations change. In contrast, irreversible inhibition results from covalent modification of the enzyme, leading to permanent inactivation.
5. Provide an example of CYP3A5 involvement in the “Clinical CYP3A4/5 Mediated Drug Interactions” section to illustrate its role and interactions in clinical contexts.
6. In line 263, the statement about "red yeast rice" inhibiting CYP3A4 is true but may be perceived as an overgeneralization. Please specify the components responsible for this inhibition, such as monacolin K, to enhance clarity and precision.
7. In line 268, correct the classification of rifampicin, which is inaccurately described as an “antipsychotic medication.” Rifampicin is an “antibiotic” primarily used to treat tuberculosis. This correction is essential for maintaining the accuracy of the drug interaction example involving rifampicin and warfarin.
8 In the “Clinical CYP3A4/5 Mediated Drug Interactions” section, it would be prudent to address how clinicians should respond to the findings discussed. Guarding practical applications and management strategies would be beneficial.
9. In line 277, the mention of "multiple subunits at its active site" may be misleading. CYP3A4 is a monomeric enzyme, meaning it consists of a single polypeptide chain and does not have multiple protein subunits or subunits like multimeric enzymes. Although CYP3A4 can form oligomers (associations of multiple enzyme molecules) in the cell, its basic functional unit is a single polypeptide. Instead, it has a large active site capable of binding multiple substrates simultaneously, which contributes to its broad substrate specificity. The description should focus on this flexibility rather than suggesting subunit composition.
10. While CYP3A5 does contribute to the metabolism of these drugs, CYP3A4 is typically the primary enzyme involved. The review should delineate the relative contributions of CYP3A5 versus CYP3A4 in these metabolic pathways and how genetic variants specifically impact these processes.
11. In line 440, The paragraph could benefit from specifying which studies or types of research have demonstrated the association between CYP3A5 polymorphisms and prostate cancer. Providing examples or citations of key studies would strengthen the argument and offer more context.

---

## Round 0.2 · accepted · Accept

All concerns of the reviewers were addressed, and the revised manuscript is acceptable now.

Reviewer 1 ·

Basic reporting

no comment

Experimental design

no comment

Validity of the findings

no comment

Additional comments

The authors address all my concerns. I suggested publishing the article in the Peer J review.

·

Basic reporting

The authors have made significant adjustments to enhance the clarity and readability of the updated manuscript.

Experimental design

No comments

Validity of the findings

The updated manuscript presents valuable findings that align well with the research process and provide significant guidance in the field.

Additional comments

The authors have addressed most of my comments as well as those of other reviewers and have made substantial updates to the manuscript. The revised version is significantly clearer and ready for publication in this journal.

Reviewer 3 ·

Basic reporting

The authors have addressed my comments. I have no additional comments. Thank you.

Experimental design

no comment

Validity of the findings

no comment